# An Orbital-Angular-Momentum- and Wavelength-Tunable 2 μm Vortex Laser

**Xinmiao Zhao, Jingliang Liu, Mingming Liu, Ruobing Li, Luan Zhang and Xinyu Chen \***

Jilin Key Laboratory of Solid Laser Technology and Application, College of Physics, Changchun University of Science and Technology, Changchun 130022, China
\* Correspondence: 2010800009@cust.edu.cn

**Abstract:** In this paper, dual tuning of orbital angular momentum (OAM) and the wavelength of a Tm:YLF vortex laser was realized by off-axis pumping and F-P etalon. The tuning of Hermite–Gaussian (HG) modes by off-axis pumping was theoretically analyzed. In the experiment, the highest 17th order $HG_{17,0}$ mode was realized by off-axis pumping. The threshold power increased from 2 to 17.51 W with the increase in off-axis distance, and the curve of threshold power vs. off-axis distance was partially consistent with the theoretical simulation analysis. The Laguerre–Gaussian (LG) modes carrying OAM were produced by mode converter, and the beam quality of LG modes was good. The phase distribution of the LG modes was verified by interference. Subsequently, an F-P etalon was inserted into the resonant cavity to tune the wavelength. Finally, the OAM tuning of the vortex beam from $LG_{1,0}$(OAM = $-1\hbar$) to $LG_{16,0}$(OAM = $-16\hbar$) was realized, and the corresponding wavelength tuning range was from 1898–1943 nm to 1898–1937 nm.

**Keywords:** Tm:YLF laser; orbital angular momentum tuning; wavelength tuning; vortex laser

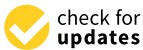



## 1. Introduction

Vortex beams with a special helical wavefront have attracted extensive research for more than three decades. The photons of these beams carry orbital angular momentum, there is a phase singularity in the center of the beams, and the light field of the beams is a hollow annular distribution. The research on vortex beams has promoted the concept of structured light. Moreover, the research is no longer limited to traditional single-singularity vortex beams. More complex structured light such as SU(2) geometric modes [1], Ince–Gaussian modes [2] and Hermite–Laguerre–Gaussian modes [3] have become the focus of research. Extensive research on vortex beams brings about important applications in many fields. In the field of optical tweezer technology, new optical tweezers and optical spanner have been developed for particle trapping and rotating through vortex beams carrying OAM [4,5]. The rotating Doppler effect of vortex beams have found important application prospects in the field of measurement [6,7]. The unique quantum properties of vortex beams have been used in the field of quantum entanglement [8]. In recent years, increasingly more people have focused on the application of vortex beams in optical communication, and especially the OAM of vortex beams has been used as a new encoding form. The simultaneous realization of wavelength-division and mode-division multiplexing can improve the multiplexing dimension to enhance the capacity of information transmission, which requires the simultaneous realization of wavelength and OAM tuning. Therefore, OAM- and wavelength-tunable vortex lasers have far-reaching significance [9–11].

The tuning of the OAM of the vortex beam can be realized by off-axis pumping [12], spatial light modulator (SLM) [13], annular laser pumping [14], etc. The common wavelength tuning methods of solid-state lasers include the use of wavelength tuning devices such as birefringent filter [15] and volume Bragg grating [16], as well as relying on crystal characteristics [17]. Wavelength- and OAM-tunable lasers have been reported in recent years. In 2018, Shen Yijie realized the tuning of HG mode and wavelength and converted

the HG mode into the corresponding LG mode [17]. In the same year, Shen Yijie realized the wavelength tuning of SU(2) geometric modes [18]. In 2017, Qiyao Liu realized the tuning of wavelength and OAM by using reflective volume Bragg grating and annular laser pumping, achieving wavelength tuning of 1643.1–1648 nm at the highest $LG_{2,0}$ mode [19]. In 2018, Sha Wang demonstrated the tuning of OAM and wavelengths by thin-film polarizer and off-axis pumping. Sha Wang achieved the $LG_{1,0}$-$LG_{14,0}$ ($1\hbar - 14\hbar$) OAM tuning, and the corresponding wavelength tuning width was from 36.2 to 14.5 nm [20].

Our team has previously studied the OAM tuning of the Tm:YLF vortex laser [21] and the wavelength tuning at the $LG_{1,0}$ mode [22]. In previous studies, off-axis pumping and F-P etalon were used as the methods of orbital angular momentum and wavelength tuning, and these two methods performed well in the two-dimensional tuning of a Tm:YLF laser. F-P etalons are rarely reported as a large-scale wavelength tuning device, as this requires extremely thin etalons. The micrometer-scale etalon can not only improve the wavelength tuning range but also greatly reduces the additional cavity loss. It also does not suppress the oscillations of high-order transverse modes such as VBG [23]. In this paper, the dual tuning of wavelength and OAM of vortex laser in the 2 µm band was studied on the basis of the previous research. Firstly, the principle of the tuning of HG modes was analyzed. Subsequently, the experimental study of off-axis pumping to achieve OAM tuning and using etalon to achieve wavelength tuning was carried out. The wavelength tuning range was optimized in comparison to previous research, and finally, wavelength tuning of nearly 40 nm at the $LG_{16,0}$ mode was achieved. This proved the good effect of this tuning method and the excellent tuning ability of the Tm:YLF laser. This work is of great significance to further promoting large-capacity optical communication.

## 2. Experiment Setup and Theoretical Analysis

### 2.1. Experiment Setup

The experimental design of this paper is shown in Figure 1a. The resonator was L-shaped, wherein M1 (HT at 792 nm, HR at 1908 nm) was a plane mirror, M2 (45° HR at 1908 nm, 45° HT at 792 nm) was a plane mirror placed at 45° compared to M1, and OC (R = 300, transmittance = 5% at 1908 nm) was a concave output mirror. A fiber-coupled 792 nm LD (core = 200 µm, NA = 0.22) was used as the pump source. The pump laser was focused to the center of the Tm:YLF crystal (size: $3 \times 3 \times 14$ mm³, Doping concentration: 3 at %) through two coupling lenses (focal length ratio = 25:50 mm). Two coupling lenses were located in a movable lens barrel, and off-axis pumping was realized by moving the lens barrel to control the position of the pumping laser. During the experiment, the crystal was placed in a heat sink connected to a water cooler to ensure that the temperature of the crystal did not exceed 20 °C. A piece of F-P etalon made of fused silica was inserted into the resonator for wavelength tuning. The thickness of the etalon was only 25 µm, and the free spectral range was 49.8 nm. The etalon was placed on a special mount that could be rotated and moved horizontally, and the wavelength was measured by a spectrometer (Yokogawa, AQ6375, Japan, 1200 nm–2400 nm). After the HG mode was generated by the off-axis pumping, the mode conversion requirements were met through the mode matching lens F1 (f = 60 mm). The HG mode satisfying the mode matching was converted into the corresponding LG mode through the mode converter. The mode converter consisted of two cylindrical lenses (f = 25 mm) placed at 45°, and the distance between the two cylindrical lenses was $\sqrt{2}f = 35.35$ mm. The transverse modes were captured by a beam profiling camera (Spiricon, Pyrocam III, USA, 13 nm–355 nm and 1.06–3000 µm). The phase distribution of the vortex beam was verified by building a Mach–Zeder interference system. The experimental setup is shown in Figure 1b. BS1 and BS2 were beam splitter and beam combiner, respectively. F2 and F3 were used to control the size of the two beams. The relative intensity of the two beams was controlled by the attenuator. The interference pattern was captured by a beam profiling camera (Spiricon, Pyrocam III, USA, 13 nm–355 nm and 1.06–3000 µm).

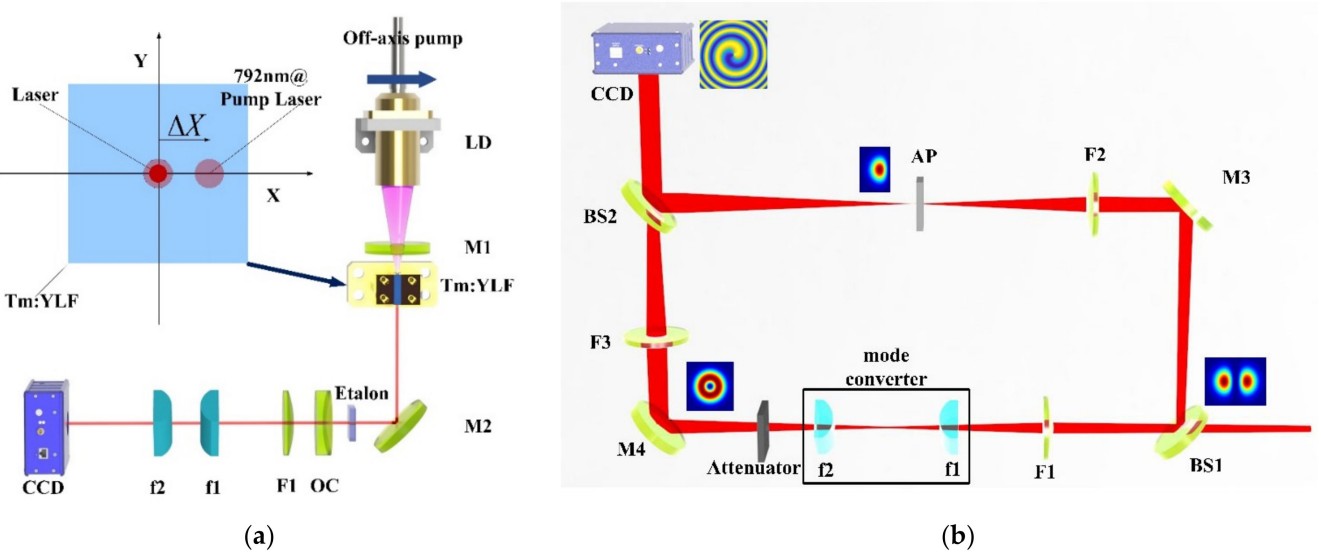

**Figure 1.** (**a**) Experimental setup of OAM- and wavelength-tunable 2 μm vortex laser. (**b**) Experimental setup of the Mach–Zeder interference system.

### 2.2. Theoretical Analysis of Off-Axis Pumping

The tuning of OAM was realized by tuning the HG mode. For off-axis pumping to achieve the HG mode order tuning, the following analysis can be made:

The threshold formula of Tm laser was

$$P_{th} = \frac{h v_p \left( \delta + 2 \Delta N^0 \sigma_{21} l \right)}{2 \sigma_{21} l \eta_{QY} \eta_a f \tau_{eff} \iiint_{crystal} r_p(x, y, z) s_{n,0}(x, y, z) dV} \tag{1}$$

In this formula, $\sigma_{21}$ is the stimulated cross-section, $l$ is the length of the Tm:YLF crystal, $\eta_{QY}$ is quantum efficiency, $\eta_a$ is the pump laser absorption efficiency, $f$ is the sum of the fractional population in the upper and lower energy levels, $\tau_{eff}$ is the effective upper state lifetime, $r_p(x, y, z)$ is the normalized spatial distribution of the pump laser, $s_{n,0}(x, y, z)$ is the normalized spatial distribution of the laser photons, $v_p$ is the pump laser frequency, $\delta$ is the sum of cavity loss and output loss, and $\Delta N^0$ is the population–inversion density when the pump power is zero. $r_p(x, y, z)$ becomes $r_p(x + \Delta x, y, z)$ when the pump laser is horizontal off-axis.

According to the formula, the threshold power vs. off-axis distance curves of six modes can be simulated, and the results are shown in Figure 2. It is known that the mode with the lowest threshold power tends to oscillate preferentially. According to Figure 2, it can be seen that the order of the lowest threshold mode gradually increased. Therefore, the whole tuning process should be a process of increasing the mode order. Furthermore, because of continuous mode change, the threshold power curve of the laser should be the same as the curve marked by the black circle in Figure 2. Moreover, the alternation between the mode curves started to become faster as the off-axis distance increased, which means an acceleration of mode changes.

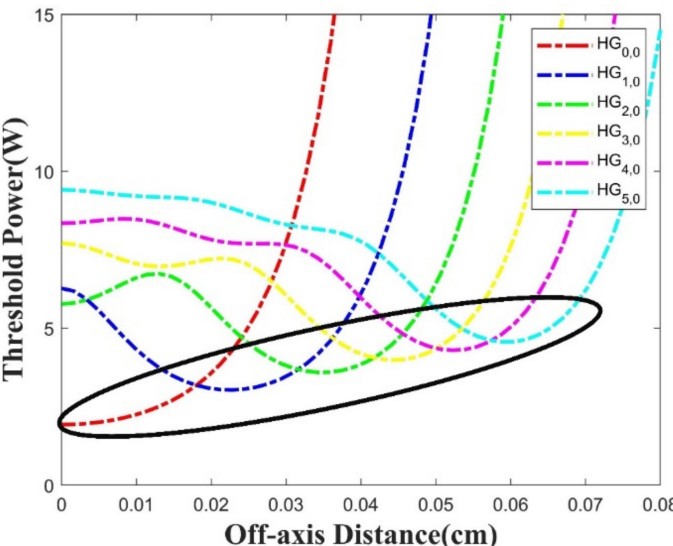

**Figure 2.** Threshold power vs. off-axis distance simulation curve of 6 modes ($\sigma_{21} = 8 \times 10^{-25} \text{m}^2$, $l = 1.4 \times 10^{-2}$m, $\tau_{eff} = 14 \times 10^{-2}$ s, $v_p = 3.787 \times 10^{14} \text{s}^{-1}$, $N_0 = 4.13 \times 10^{26}$ m$^{-3}$, $n_{QY} = 1.81$, $n_a = 0.907$, $f = 0.319$, $\delta = 0.065$).

## 3. Experimental Results and Discussion

Firstly, the HG mode tuned by off-axis pumping was studied. By controlling the off-axis distance, HG modes of different orders were generated. The HG mode and threshold power at different off-axis distances are shown in Figure 3a. With the increase in the off-axis distance, the threshold power increased from 2 W to the highest 17.51 W, and the order of HG mode increased to the highest HG$_{17,0}$. The curve of threshold power vs. off-axis distance partially showed the shape of multi-segment curve stitching, which was similar to the theoretical simulation analysis. Moreover, the turning points of the curve were located in the interval of the mode change. The reason why multi-segment curve stitching was partially shown was that the measurement accuracy cannot show the full picture of the curve due to the acceleration of the mode change. Afterwards, the conversion from HG mode to LG mode was completed by the mode converter. The conversion result is shown in Figure 3a. The lobed HG modes were converted into the ring-shaped LG modes. The beam quality of the LG modes was measured by beam profiling camera. The theoretical $M^2$ of the LG mode in two directions is $M_X^2 = M_Y^2 = 2p + l + 1$. The experimental $M^2$ of the LG modes are shown in Figure 3b. The beam quality decreased with the increase in the mode order, and the highest deviation was 1.16 from the theoretical value at the LG$_{17,0}$ Y direction. However, considering that the theoretical $M^2$ value was high, the deviation was only 6.44%. Therefore, it can be considered that the beam quality was good. The phase distribution of the vortex beam was verified by interference experiments, and the experimental results are shown in Figure 3c. It can be seen that the interference pattern was spiral interference fringes. These spiral fringes were the embodiment of the spiral phase distribution of the measured beam.

On the basis of the OAM tuning experiment, a F-P etalon was inserted into the resonator to carry out the two-dimensional tuning experiment. If off-axis pumping is tuning the gain for different transverse modes, then the F-P etalon is the gain for different wavelengths. These two tuning methods are more independent of each other. The dual tuning method was used to achieve wavelength tuning while maintaining the output of LG$_{n,0}$ mode. The tuning results are shown in Figure 4. At 6.15 W pump power, the maximum wavelength tuning range was achieved from 1898 to 1943 nm, while the LG$_{1,0}$ mode (OAM = $-1\hbar$) was maintained. As shown by the blue arrow in Figure 4a, the tuning process of the wavelength was from 1937 to 1898 nm, and then abruptly to 1943 nm, and following this to 1937 nm. At 25.31 W pump power, the minimum wavelength tuning range was achieved from 1898 to 1937 nm, while the LG$_{16,0}$ mode (OAM = $-16\hbar$) was

maintained. As shown in Figure 4a, the wavelength was continuously tuned from 1937 to 1898 nm. The wavelength tuning range for other modes was between $LG_{1,0}$ mode and $LG_{16,0}$ mode. The reason why the wavelength tuning width decreased with the increase in the LG mode order was that the higher resonator loss of the higher-order modes makes it difficult for some Tm:YLF uncommon bands to oscillate. Moreover, because the insertion of the etalon increased the resonator loss, the order that can be achieved by off-axis pump tuning was reduced from 17 to 16. It should be especially mentioned that the wavelength tuning range from 1921 to 1937 nm was achieved by rotating the etalon in the previous study [22], which was much smaller than the free spectral range of the etalon (49.8 nm). The main reason limiting the wavelength tuning range was the fact that the clear aperture of the etalon was small, and the rotation axis of the etalon was not the centerline of the etalon, causing the laser to no longer pass through the center of the etalon. As the rotation angle of the etalon increased, the actual clear aperture was greatly reduced, and the laser was difficult to oscillate. Finally, the wavelength tuning range was greatly increased by moving the etalon horizontally while rotating the etalon. Compared to previous double-tuning studies, our mode tuning range as well as wavelength tuning range were improved. This was mainly due to the extremely thin etalon, which had a large free spectral range and low additional cavity losses.

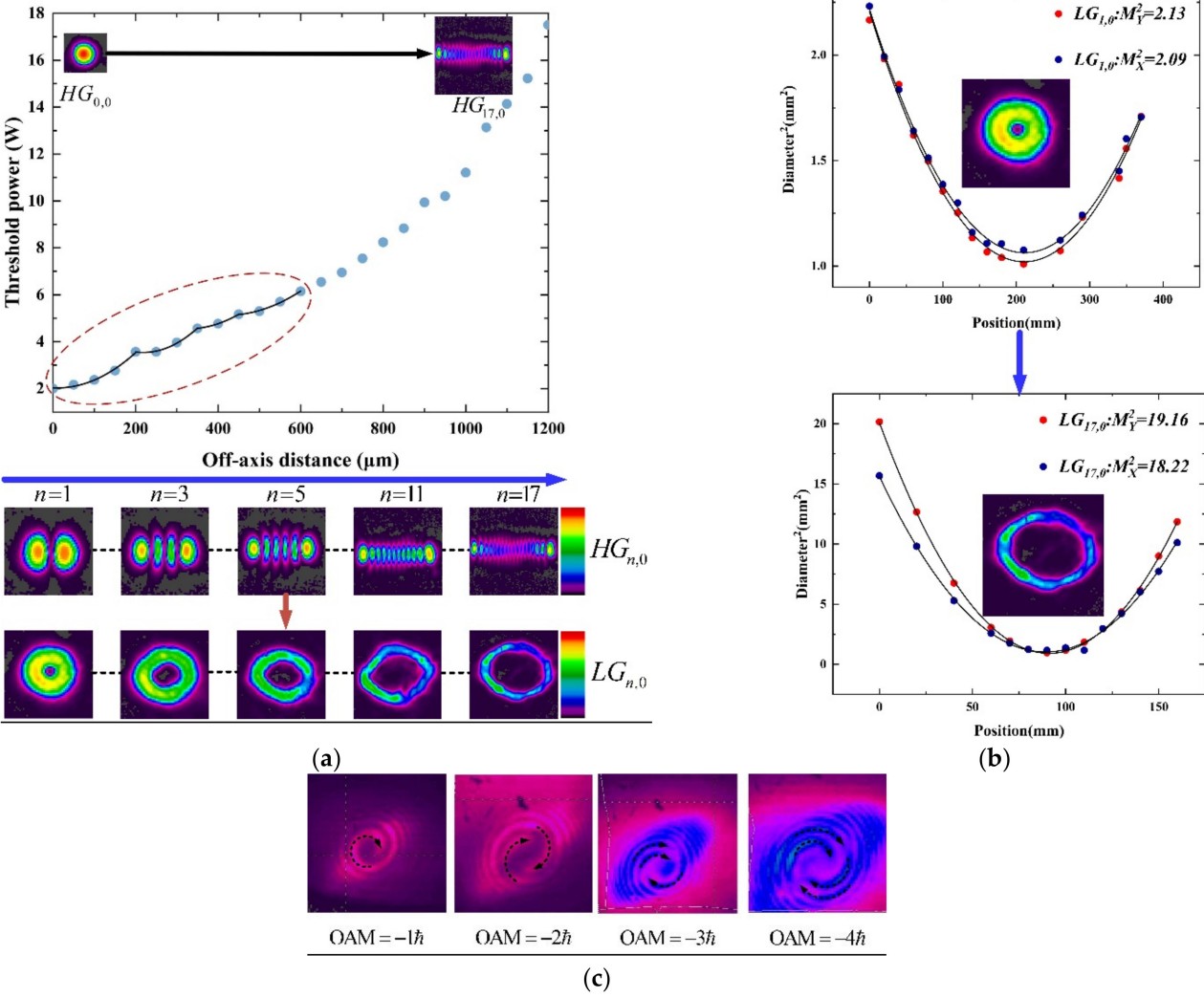

**Figure 3.** (**a**) Threshold power of the laser during off-axis processes and measured $HG_{n,0}$ modes and $LG_{n,0}$ modes. (**b**) Beam quality of LG modes. (**c**) The interference pattern of the vortex beam with the spherical wave.

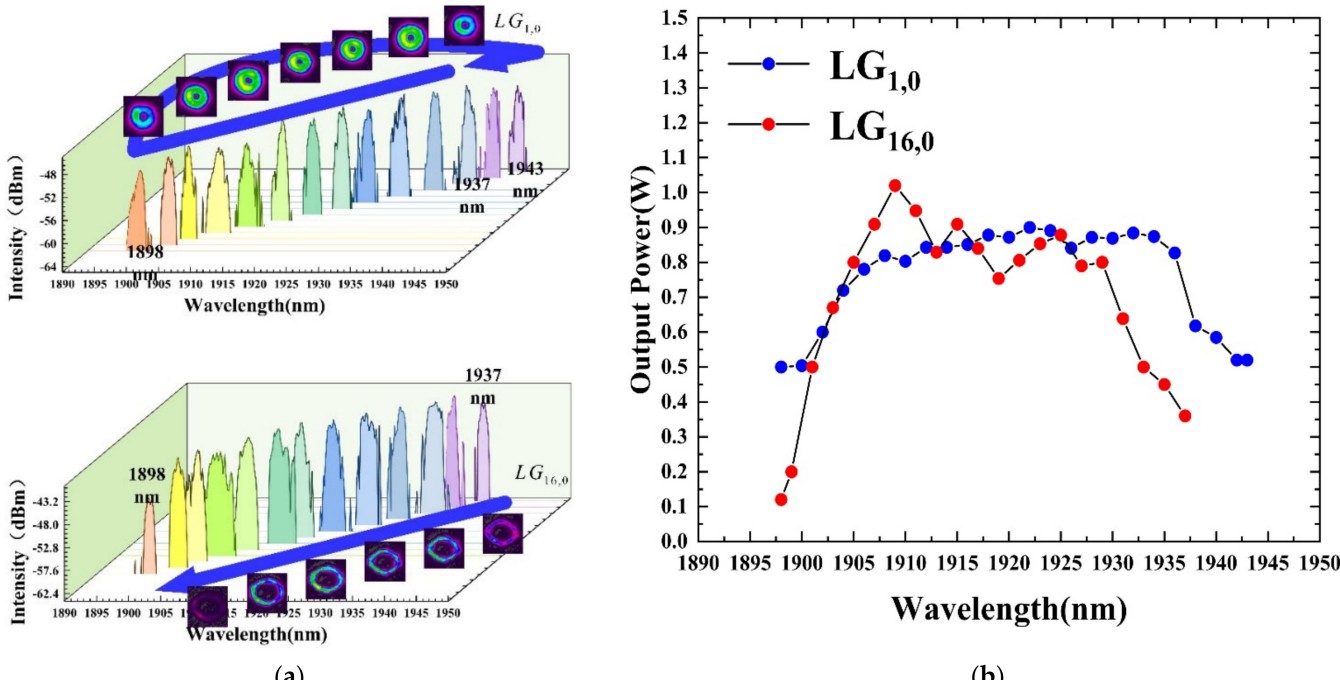

**Figure 4.** (**a**) Wavelength tuning of $LG_{1,0}$ mode and $LG_{16,0}$ mode. (**b**) Output power curve for different wavelengths.

## 4. Conclusions

In this paper, a Tm:YLF vortex laser with tunable wavelength and OAM was demonstrated. Through the off-axis pumping experiment, the HG mode output from the $HG_{0,0}$ to the highest $HG_{17,0}$ was realized. During the off-axis pumping process, the threshold power increased from 2 to 17.51 W, and the measured curve of threshold power vs. off-axis distance partially showed the same results as the simulation results. The mode converter composed of two cylindrical lenses realized the conversion from HG modes to LG modes, and the beam quality of LG modes was good. The phase distribution of the converted LG mode was verified by interference with a spherical wave. Subsequently, on the basis of OAM tuning, the dual tuning of OAM and wavelength was realized by rotating the etalon. Finally, the OAM tuning of the vortex beam from $LG_{1,0}$ (OAM = $-1\hbar$) to $LG_{16,0}$ (OAM = $-16\hbar$) was realized, and the corresponding wavelength tuning range was from 1898–1943 nm to 1898–1937 nm. In this paper, off-axis pumping and F-P etalon were used to show excellent multi-dimensional tuning ability in Tm:YLF laser. This technology is also applicable to other solid-state lasers, and in-depth study of multi-dimensional tuning is of great significance for large-capacity optical communication.

**Author Contributions:** Conceptualization, X.Z. and J.L.; methodology, X.Z.; software, X.Z. and R.L.; validation, X.Z. and J.L.; formal analysis, X.Z.; investigation, X.Z. and L.Z.; resources, X.Z.; data curation, X.Z.; writing—original draft preparation, X.Z.; writing—review and editing, X.Z.; visualization, M.L.; supervision, X.C.; project administration, X.C.; funding acquisition, J.L. All authors have read and agreed to the published version of the manuscript.

**Funding:** This research was supported by Project of Jilin Scientific and Technological Development Program (YDZJ202101ZYTS031).

**Institutional Review Board Statement:** Not applicable.

**Informed Consent Statement:** Not applicable.

**Data Availability Statement:** The data that support the findings of this study are available from the corresponding author upon reasonable request.

**Conflicts of Interest:** The authors declare no conflict of interest.

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
