# Peer review of "An Orbital-Angular-Momentum- and Wavelength-Tunable 2 μm Vortex Laser"

_photonics, doi:10.3390/photonics9120926_

Round 1
Reviewer 1 Report
This is a very anticipated work, tunable structured light in additional dimensions is a very hot topic, and the vortex- and wavelength-tunable laser at 2μm is a previous technological gap, I am happy to see the authors fill this gap. I can recommend publication after minor revisions:
(1) There is syntactical error in title for the use of hyphen, it should be "Orbital-angular-momentum- and wavelength-tunable" or "Orbital angular momentum and wavelength tunable"
(2) The author should introduce more background of optical vortices and higher-dimensional structured light, please refer to the related review articles recently in LSA.
(3) Some prior OAM- and wavelength-tunable laser works were missed, e.g. AO 57 (32), 9543-9549 (2018)
(4) Add colorbar to the plots of intensity patterns.
(5) Add experimental interference patterns to verify the exist of OAM.
Author Response
Thank you for your comments concerning our manuscript entitled “Orbital-angular-momentum-and wavelength-tunable 2μm vor-tex laser”. Those comments are all valuable and very helpful for revising and improving our paper, as well as the important guiding significance to our researches. We have studied comments carefully and have made correction which we hope meet with approval.
Point 1: There is syntactical error in title for the use of hyphen, it should be "Orbital-angular-momentum- and wavelength-tunable" or "Orbital angular momentum and wavelength tunable".
Response 1: Thank you for pointing out this problem in our manuscript. We have revised the title of the manuscript.
Point 2: The author should introduce more background of optical vortices and higher-dimensional structured light, please refer to the related review articles recently in LSA.
Response 2: Thank you for the above suggestion, we have enriched the introduction.
Point 3: Some prior OAM- and wavelength-tunable laser works were missed, e.g. AO 57 (32), 9543-9549 (2018).
Response 3: Thank you for the above suggestion, We have introduced this article in the revised manuscript.
Point 4: Add colorbar to the plots of intensity patterns.
Response 4: As Reviewer suggested that we have added colorbar to the plots of intensity patterns.
Point 5: Add experimental interference patterns to verify the exist of OAM.
Response 5: As Reviewer suggested that we have added experimental interference patterns.
Reviewer 2 Report
The manuscript addresses the Tm:YLF vortex laser with tunable wavelength and orbital angular momentum (OAM). It is shown that through the off-axis pumping experiment, the Hermite-Gaussian (HG) mode output from the HG(0,0) to the highest HG(17,0) has been realized, and the Laguerre-Gaussian (LG) modes carrying OAM has been produced via the conversion from HG modes. Further by F-P etalon together with off-axis pumping, dual tuning of OAM and wavelength of Tm:YLF vortex laser has been achieved. The investigations may have potential applications on high-performance optical communication.
I have several remarks as following.
1) In the simulation on threshold power as a function of the off-axis distance, the authors should provide all the parameters either in the main text or in the caption of Fig.1(b).
2) In the experiments, the authors should measure both intensity and phase distribution of the beam and provide the data, for example, in Figs. 2 and 3.
3) It is better to add the discussion on the conversion efficiency from HG modes and LG modes.
Besides, the citation of references in the introduction part is not formal, and the English should be polished.
I think this manuscript may present the interesting experimental results, and it can be published in Photonics after major revision.
Author Response
Thank you for your comments concerning our manuscript entitled “Orbital-angular-momentum-and wavelength-tunable 2μm vor-tex laser”. Those comments are all valuable and very helpful for revising and improving our paper, as well as the important guiding significance to our researches. We have studied comments carefully and have made correction which we hope meet with approval.
Point 1: In the simulation on threshold power as a function of the off-axis distance, the authors should provide all the parameters either in the main text or in the caption of Fig.1(b).
Response 1: Thank you for the above suggestion. We have provide all the parameters in the caption.
Point 2: In the experiments, the authors should measure both intensity and phase distribution of the beam and provide the data, for example, in Figs. 2 and 3.
Response 2: As Reviewer suggested that we have added experimental interference patterns.
Point 3: It is better to add the discussion on the conversion efficiency from HG modes and LG modes.
Response 3: Thank you for the above suggestion. In fact, there is almost no loss in the conversion. Conversion effect similar to beam through ordinary uncoated lens.
Point 4: The citation of references in the introduction part is not formal, and the English should be polished.
Response 4: Thank you for pointing out this problem in our manuscript. We have revised the manuscript carefully and tried to avoid any grammar or syntax error.
Reviewer 3 Report
In this manuscript, the authors present the way to realize dual tuning of OAM and wavelength of vortex laser, and experimentally demonstrate the design. This work may be applicable in optical communication. However, it needs to be revised before it could be published. My opinions are as follows.
1. Please clarify the novelty of this work as it is an interesting idea built directly on the authors’ previous works. Also, please give more information on the advantages of this work over literatures (e.g., Refs. [16] and [17]).
2. Please give more theoretical explanation on how to achieve large-scale wavelength tuning simultaneously with OAM tuning.
3. Caption of Figure 2 is incorrect.
4. The language needs improving throughout the text.
Author Response
Thank you for your comments concerning our manuscript entitled “Orbital-angular-momentum-and wavelength-tunable 2μm vor-tex laser”. Those comments are all valuable and very helpful for revising and improving our paper, as well as the important guiding significance to our researches. We have studied comments carefully and have made correction which we hope meet with approval.
Point 1: Please clarify the novelty of this work as it is an interesting idea built directly on the authors’ previous works. Also, please give more information on the advantages of this work over literatures (e.g., Refs. [16] and [17])
Response 1: Thank you for the above suggestion. We have stated more about the novelty and advantages in the revised manuscript.
Point 2: Please give more theoretical explanation on how to achieve large-scale wavelength tuning simultaneously with OAM tuning.
Response 2: Thank you for the above suggestion. These two tuning methods are more like two independent tuning methods. Off-axis pumping is to achieve gain control for the transverse mode by controlling the overlap between different modes and pump laser. The gain of different wavelengths is controlled by the spectral response curve of etalon. The connection so far is that the higher-order transverse modes have higher cavity losses affecting the wavelength tuning range.
Point 3: Caption of Figure 2 is incorrect.
Response 3: Thank you for pointing out this problem in our manuscript. We have revised the Caption.
Point 4: The language needs improving throughout the text.
Response 4: As Reviewer suggested that we have revised the manuscript carefully and tried to avoid any grammar or syntax error.
Round 2
Reviewer 2 Report
The revised manuscript has been improved. As I mentioned before, this work presents an interesting experimental result. I think it can be considered to be published in Photonics.
Reviewer 3 Report
There is no more question from this reviewer.